# Decreased Levels of Microfibril-Associated Glycoprotein (MAGP)-1 in Patients with Colon Cancer and Obesity Are Associated with Changes in Extracellular Matrix Remodelling

**DOI:** 10.3390/ijms22168485

**Published:** 2021-08-06

**Authors:** Iranzu Gómez de Segura, Patricia Ahechu, Javier Gómez-Ambrosi, Amaia Rodríguez, Beatriz Ramírez, Sara Becerril, Xabier Unamuno, Amaia Mentxaka, Jorge Baixauli, Víctor Valentí, Rafael Moncada, Camilo Silva, Gema Frühbeck, Victoria Catalán

**Affiliations:** 1Metabolic Research Laboratory, Clínica Universidad de Navarra, 31008 Pamplona, Spain; igomezdeseg@alumni.unav.es (I.G.d.S.); jagomez@unav.es (J.G.-A.); arodmur@unav.es (A.R.); bearamirez@unav.es (B.R.); sbecman@unav.es (S.B.); xunamuno@u.es (X.U.); amentxaka@alumni.unav.es (A.M.); 2Department of Surgery, Clínica Universidad de Navarra, 31008 Pamplona, Spain; pahechu@unav.es (P.A.); jbaixauli@unav.es (J.B.); vvalenti@unav.es (V.V.); 3CIBEROBN, Instituto de Salud Carlos III, 31008 Pamplona, Spain; rmoncada@unav.es (R.M.); csilvafr@unav.es (C.S.); 4Obesity and Adipobiology Group, IdiSNA, 31008 Pamplona, Spain; 5Department of Anesthesia, Clínica Universidad de Navarra, 31008 Pamplona, Spain; 6Department of Endocrinology & Nutrition, Clínica Universidad de Navarra, 31008 Pamplona, Spain

**Keywords:** obesity, colon cancer, adipose tissue, inflammation, extracellular matrix remodelling, MAGP-1, TGF-β

## Abstract

Objective: The protein microfibril-associated glycoprotein (MAGP)-1 constitutes a crucial extracellular matrix protein. We aimed to determine its impact on visceral adipose tissue (VAT) remodelling during obesity-associated colon cancer (CC). Methods: Samples obtained from 79 subjects (29 normoponderal (NP) (17 with CC) and 50 patients with obesity (OB) (19 with CC)) were used in the study. Circulating concentrations of MAGP-1 and its gene expression levels (*MFAP2*) in VAT were analysed. The impact of inflammation-related factors and adipocyte-conditioned media (ACM) on *MFAP2* mRNA levels in colon adenocarcinoma HT-29 cells were further analysed. The effects of MAGP-1 in the expression of genes involved in the extracellular matrix (ECM) remodelling and tumorigenesis in HT-29 cells was also explored. Results: Obesity (*p* < 0.01) and CC (*p* < 0.001) significantly decreased *MFAP2* gene expression levels in VAT whereas an opposite trend in *TGFB1* mRNA levels was observed. Increased mRNA levels of *MFAP2* after the stimulation of HT-29 cells with lipopolysaccharide (LPS) (*p* < 0.01) and interleukin (IL)-4 (*p* < 0.01) together with a downregulation (*p* < 0.05) after hypoxia mimicked by CoCl_2_ treatment was observed. MAGP-1 treatment significantly enhanced the mRNA levels of the ECM-remodelling genes collagen type 6 α3 chain (*COL6A3)* (*p* < 0.05), decorin (*DCN*) (*p* < 0.01), osteopontin (*SPP1*) (*p* < 0.05) and *TGFB1* (*p* < 0.05). Furthermore, MAGP-1 significantly reduced (*p* < 0.05) the gene expression levels of prostaglandin-endoperoxide synthase 2 (*COX2*/*PTGS2*), a key gene controlling cell proliferation, growth and adhesion in CC. Interestingly, a significant decrease (*p* < 0.01) in the mRNA levels of *MFAP2* in HT-29 cells preincubated with ACM from volunteers with obesity compared with control media was observed. **Conclusion:** The decreased levels of MAGP-1 in patients with obesity and CC together with its capacity to modulate key genes involved in ECM remodelling and tumorigenesis suggest MAGP-1 as a link between AT excess and obesity-associated CC development.

## 1. Introduction

Colon cancer (CC) is the second leading cause of death in developed societies [1]. More than one-half of all causes of deaths related to cancer are attributable to modifiable risk factors including smoking, alcohol consumption, lack of physical activity and excess body weight with, specifically, 8% of all cancer deaths being associated to obesity [2]. Obesity, defined as an excess of adipose tissue (AT), is a concern worldwide due to its increased prevalence together with its well-known association with a cluster of chronic diseases including type 2 diabetes, cardiovascular diseases and cancer among others [3,4,5,6]. Moreover, obesity might result in delayed cancer diagnosis due to the hemodilution of several tumor biomarkers, technical difficulties and reduced image quality in medical imaging as well as lower participation in cancer screening programs [7]. Growing evidence indicates that obesity promotes CC not only by the tumour-promoting effects of systemically dysregulated adipokines and metabolic mediators, but also by local progression enhancement through chronic low-grade inflammation [4,5,8,9] and the extracellular matrix (ECM) remodelling impairment of the AT [9,10,11]. In this sense, the ECM of the AT is now recognized as a dynamic regulator of cellular processes providing structural support to the surrounding cells and, thus, playing a crucial role in the biological function of different organs [12,13]. However, the dysfunctional AT in the obese state is associated with an increased deposition of ECM components and the activation of ECM receptor pathways promoting a dysregulation of inflammation and fibrosis processes [12] and favouring the development of CC [10].

The ECM of AT is composed of a milieu of basement membrane components, fibrillar and non-fibrillar collagens, fibronectin, microfibrils and different mechanotransduction proteins [13]. Among these components, the microfibril superstructures exhibit a significant relevance since they provide structural support and regulate the bioavailability of growth factors, particularly members of the transforming growth factor (TGF)-β superfamily [14]. Microfibrils are formed by structural proteins named fibrillins and microfibril-associated proteins (MAGPs). An important member of the MAGP family is the microfibril-associated glycoprotein-1 (MAGP-1), a small protein of 21 kDa constituent of most vertebrate microfibrils coded by the microfibril-associated protein 2 (*MFAP2*) gene and located on the human chromosome 1p31 [15,16]. MAGP-1 interacts noncovalently with fibrillin-1 and the active form of TGF-β by its N-terminal position [17]. In this regard, MAGP-1 exhibits dual roles in AT: (i) to provide mechanical stability and limited elasticity to AT through the interaction with fibrillin-1 to form a functional fiber and (ii) to regulate crucial metabolic pathways including fibrosis, inflammation and thermogenesis by regulating the bioavailability of the active forms of TGF-β [18,19,20].

MAGP-1 has been linked with complex phenotypes in multiple organ systems [18,19,20,21]. In this sense, *Mfap2*-deficient mice exhibited a phenotype of increased adiposity and impaired thermoregulation consistent with predisposition to metabolic dysfunction and increased TGF-β activity [22,23,24]. The treatment of *Mfap2* knockout mice with antibodies neutralizing TGF-β prevented increased adiposity, proposing its role in the regulation of TGF-β in obesity and its associated comorbidities [17]. Thus, MAGP-1 has been suggested as a protective mechanism against the effects of metabolic stress, with a lack of MAGP-1 predisposing to metabolic dysfunction [22].

Different studies have also implicated MAGP-1 in tumour progression. In this line, MAGP-1 levels are known to be overexpressed in gastric cancer [25] and multiple myeloma [26]. The downregulation of MAGP-1 has been shown to inhibit the migration and invasion of gastric cancer cells with no impact on their capacity of proliferation [25]. Interestingly, MAGP-1 has been also implicated in the epithelial-mesenchymal transition (EMT) due to its interaction with TGF-β, SMAD2/3 and WNT/NOTCH pathways [27,28]. Increased levels of TGF-β in obesity have been widely associated with inflammation and fibrosis creating a favourable microenvironment for CC development. Reportedly, an excessive production and/or activation of TGF-β promotes CC progression and predicts adverse outcomes in patients with obesity [29].

To our knowledge, few studies have evaluated the role of MAGP-1 in human obesity-associated CC. Since MAGP-1 is reported as an important ECM component of the AT with a role in cancer progression, we hypothesized that dysregulated levels of MAGP-1 in obesity may function as a link between AT excess and CC development. Therefore, the aim of the present study was to investigate whether obesity can influence the circulating concentrations of MAGP-1 and its gene expression levels in patients with CC promoting a microenvironment favourable for tumour growth. We determined the circulating concentrations as well as gene expression levels of *MFAP2* and *TGFB1* in the visceral AT (VAT) from patients with and without CC. To gain insight into the molecular mechanisms involved, the effect of different inflammation-related factors on the expression levels of *MFAP2* and *TGFB* as well as in key ECM remodelling factors in colon adenocarcinoma HT-29 cells was explored. We also evaluated whether MAGP-1 itself can modulate the inflammatory response in the HT-29 cell line. Finally, the impact of the adipocyte-conditioned medium (ACM) obtained from patients with obesity on the expression of *MFAP2* in HT-29 cells was further analyzed.

## 2. Results

### 2.1. Obesity and Colon Cancer Decrease Circulating Concentrations of MAGP-1 and Its Gene Expression Levels in VAT

Baseline characteristics of the study sample to analyse the effect of obesity and CC are shown in Table 1. As expected, patients from groups of patients with OB exhibited an increase (*p* < 0.001) in all anthropometric measurements compared to the NP volunteers but no differences were detected between patients with or without CC. Patients with CC were older (*p* < 0.001) compared to those without CC. Patients with OB and CC showed increased (*p* < 0.001) levels of CRP compared with NP and patients with OB but without CC. Moreover, fibrinogen concentrations were higher (*p* < 0.01) in patients with OB and CC compared with patients with OB without CC. Circulating concentrations of interleukin (IL)-6 were significantly increased due to obesity (*p* = 0.002) and CC (*p* < 0.001). We also found that obesity (*p* = 0.006) and CC (*p* = 0.012) increased plasma levels of chitinase 3-like 1 (YKL-40), a marker that contributes to chronic inflammation and oncogenic transformation. In the same line, circulating levels of osteopontin (OPN) were upregulated (*p* < 0.001) by OB and CC. We also found that obesity increased (*p* = 0.038) plasma levels of vascular endothelial growth factor A (VEGFA). Oppositely, circulating levels of the anti-inflammatory cytokine IL-4 showed lower levels due to obesity (*p* = 0.038) and CC (*p* = 0.046) and IL-13 concentrations were reduced (*p* = 0.016) in patients with obesity, but no effect regarding the presence or not of CC was found. No differences were found in the global white blood cells count regarding obesity, but patients with CC exhibited an elevated number of leucocytes (*p* < 0.05) compared with those without CC.

Circulating MAGP-1 concentrations were reduced due to CC (*p* = 0.016) but no differences were found regarding obesity (*p* = 0.276) (Figure 1A). No significant differences were found between experimental groups in the circulating concentrations of TGF-β (Figure 1B). Based on its relevance in obesity-associated inflammation and colon carcinogenesis, the VAT was selected to study *MFAP2* and *TGFB1* gene expression levels. Significant differences in *MFAP2* gene expression levels were observed, being significantly decreased due to obesity (*p* < 0.010) and CC (*p* < 0.001) (Figure 1C). Oppositely, a tendency towards an increased gene expression levels of *TGFB1* in the VAT from patients with OB and CC was found although differences were only significant for CC (*p* < 0.05) (Figure 1D). We also analysed the influence of OB and CC in the expression levels of the ECM remodelling factor collagen 6A3 (*COL6A3*) and decorin (*DCN*) (Appendix A). Gene expression levels of *COL6A3* were significantly increased (*p* = 0.019) in the VAT from patients with CC. Although *DCN* expression tended to be higher due to CC, differences did not reach statistical significance.

### 2.2. Role of Inflammation-Related Factors and Hypoxia in MFAP2 mRNA Levels in Colon Adenocarcinoma HT-29 Cells

Since it is widely recognized that chronic low-grade inflammation is a cardinal feature of CC local progression enhancement, we next evaluated whether well-known inflammation-related factors dysregulated in obesity influence *MFAP2* expression in HT-29 cells. Unexpectedly, the stimulation with LPS resulted in increased (*p* < 0.01) *MFAP2* gene expressions levels (Figure 2A), but no significant differences were observed on *MFAP2* expression after TNF-α treatment (Figure 2B). Upregulated *MFAP2* mRNA levels after the treatment with the anti-inflammatory factor IL-4 was found (*p* < 0.01) (Figure 2C) while no significant differences after IL-13 treatment were observed (Figure 2D). HT-29 cells under hypoxia, achieved by the incubation with the divalent transition-metal ion cobalt (CoCl_2_), exhibited downregulated (*p* < 0.05) *MFAP2* expression levels at the higher concentration (Figure 2E). TGF-β treatment showed an increase in the mRNA levels of *MFAP2*, in accordance with its known binding capacity (Figure 2F).

We also analyzed the effects of these inflammation-related factors in the gene expression levels of COL6A3 and DCN, two ECM proteins strongly associated with MAGP-1 (Appendix A). No differences were found in the gene expression levels of COL6A3 and DCN after the treatment of HT-29 with LPS and TNF-α. Although no differences were found in the mRNA levels of COL6A3 after the stimulation of HT-29 cells with the anti-inflammatory factors IL-4 and IL-13, the expression levels of DCN were significantly downregulated (*p* < 0.05) after IL-13 treatment.

### 2.3. MAGP-1 Regulates the Expression of ECM- and Tumorigenesis-Related Factors in HT-29 Cells

MAGP-1 is an important component of the ECM of AT that plays a crucial role in ECM remodelling by its binding to active TGF-β. MAGP-1 treatment significantly enhanced the mRNA levels of the ECM-remodelling genes *COL6A3* (*p* < 0.05), *DCN* (*p* < 0.01), *SPP1* (*p* < 0.05) and *TGFB1* (*p* < 0.05) (Figure 3A,B). We also analyzed the impact of MAGP-1 on the control of the expression of specific genes regulating cell proliferation, growth and adhesion in CC. In this regard, MAGP-1 treatment reduced (*p* < 0.05) gene expression levels of *COX2*. However, no significant differences in the expression levels of *CTNNB1* and *MUC2* were found after treatment with MAGP-1 (Figure 3B).

### 2.4. Adipocyte-Conditioned Media Downregulates Gene Expression Levels of MFAP2 in HT-29 Cells

To gain insight into the molecular mechanisms behind the adipocyte-cancer cell crosstalk, we evaluated the effect of ACM on the mRNA levels of *MFAP2* in HT-29 cells. Interestingly, we found a significant decrease (*p* < 0.01) in the expression levels of *MFAP2* after incubation with the ACM obtained from volunteers with obesity (Figure 4).

## 3. Discussion

MAGP-1 constitutes a secreted signal peptide initially identified as a key component of extracellular microfibrils [15,16]. MAGP-1 shows a significant role in AT, protecting against obesity and metabolic dysfunction through the regulation of TGF-β in animal models [22]. In addition, a significant involvement of MAGP-1 in tumour progression has also been demonstrated [25,26,30,31,32]. However, the possible role of MAGP-1 in the remodelling of the AT as a mechanism linking obesity and CC remains unclear. In this regard, the present study was designed to determine the function of MAGP-1 in obesity-associated CC. The major findings of our investigation are that: (i) obesity and CC reduce circulating and gene expression levels of MAGP-1; (ii) inflammation-related factors and hypoxia regulate *MFAP2* mRNA levels in HT-29 colon cancer cells; (iii) MAGP-1 modulates the expression of ECM-related and tumorigenesis genes in HT-29 cells and (iv) the ACM obtained from patients with obesity downregulates the gene expression levels of *MFAP2* in HT-29 cells.

To our knowledge, the present study is the first describing decreased circulating concentrations of MAGP-1 in patients with obesity and CC. Unexpectedly, neither obesity nor CC significantly changed circulating TGF-β concentrations between the experimental groups. Different studies have described dysregulated MAGP-1 levels in diverse cancers underlining its implication in tumour progression, particularly in the processes of epithelial mesenchymal transition (EMT) due to its interaction with TGF-β [27]. In this sense, opposite gene expression levels of *MFAP2* have been described in different tumour types as being upregulated in gastric [25], thyroid [33] and hepatocellular [34] carcinomas and downregulated in prostate carcinoma and paragangliomas [35].

MAGP-1 transcript levels have been detected in white, beige and brown AT depots from mice with its expression levels being increased in the epididymal AT of obese mice [17]. Similarly, *MFAP2* expression levels in human subcutaneous AT have been shown to be positively correlated with BMI [22]. In the present study, we report significantly decreased *MFAP2* gene expression levels in the VAT due to obesity and CC together with an opposite trend in the gene expression levels of *TGFB1*. Considering the high affinity interaction between MAGP-1 and TGF-β, the inverse regulation of their gene expression levels may indicate that the decreased levels of MAGP-1 coexist with higher free active TGF-β, prompting its interaction with the surrounding cells in the ECM of VAT. Thus, the role of MAGP-1 in inhibiting the inflammatory response may result in preventing the presence of an excessive concentration of available active TGF-β. This finding suggests that the regulation of TGF-β by MAGP-1 may be a protective mechanism against the effects of metabolic stress with its absence predisposing to metabolic dysfunction. As reviewed by Tan et al. [36], the excess of TGF-β in obesity facilitates the development of metabolic syndrome by promoting inflammation and fibrosis, which impairs adipose tissue functionality by limiting adipocyte expansion. In mice, the inhibition of TGF-β has been shown as a protective mechanism against diet-induced obesity and diabetes [37]. Moreover, VAT is considered the most pathogenic AT and the most relevant in obesity-associated inflammation and colon carcinogenesis. In this line, an excessive production and/or activation of TGF-β promotes CC progression with TGF-β circulating levels also predicting adverse outcomes in patients with CC [38]. Taking together, MAGP-1 and TGF-β misbalance in the VAT from patients with obesity may participate in local inflammation and ECM remodelling, fostering obesity-related CC development. Although different tissues express and secrete MAGP-1, its decreased expression in VAT together with the reduced circulating concentration in patients with CC suggests that decreased systemic plasma levels may be in part the result of the reduced expression in VAT. In this context, obesity and CC also increased the levels of relevant inflammation and ECM remodelling factors including IL-6, OPN, YKL-40 and VEGFA. No influence of pathological features of CC in MAGP-1/*MFAP2* and TGF-β1/*TGFB1* expression have been observed in our study. However, the extracellular expression of *TGFB1* has been associated with ER-positive status [39,40] being also more frequent in tumors with lymph node metastases [41]. Moreover, a different function of TGF-β1 signaling between non-metastatic and metastatic prostate cancer has been previously described [42].

Inflammation is considered a crucial factor driving carcinogenesis and, therefore, we analysed the role of the inflammation-related molecules in *MFAP2* expression. In accordance with its anti-inflammatory function [43], increased levels of *MFPA2* mRNA were found in HT-29 cells after IL-4 treatment. *MFAP2* mRNA level upregulation may be a mechanism triggered to bind excess TGF-β to control inflammation. In this regard, HT-29 cells under hypoxia entailed a significant downregulation of *MFAP2* expression levels, pointing to hypoxia as another mechanism implicated in the regulation of *MFAP2*. Surprisingly, the stimulation of HT-29 cells with the well-known inflammatory factor LPS resulted in an increased *MFAP2* gene expression, whereas no significant changes were observed on *MFAP2* expression after TNF-α treatment. While LPS is a potent exogenous inflammatory factor, TNF-α is a less intense, intrinsic and receptor-mediated inflammatory factor. LPS-mediated inflammation pathways may be involved in MAGP-1 transduction signals, reflecting an initial compensatory protective mechanism to control increased TGF-β levels in response to pathologic fat accumulation and its related inflammation. The treatment of HT-29 cells with TGF-β initially increased the gene expression levels of *MFAP2*, but higher concentrations of TGF-β involved a gradual downregulation in its expression levels. Given the capacity of MAGP-1 to sequester active TGF-β in the extracellular matrix of adipose tissue, we hypothesize that *MFAP2* expression levels may be upregulated as a protective response to the increased concentrations of TGF-β, with its higher concentrations exceeding the *MFAP2* response capacity and inducing an exhausted compensatory mechanism.

MAGP-1 is anchored to the bead regions of fibrillin-containing microfibrils by its C-terminal domain and to surrounding matrix proteins including active forms of TGF-β1, TGF-β2, bone morphogenetic protein (BMP)-2, -4 and -7 [17], α3 chain of the collagen 6, decorin, biglycan and fibrillin-1 [15] by its N-terminal domain. In this regard, we have established that increasing concentrations of MAGP-1 significantly enhance the gene expression levels of the ECM-remodelling genes *COL6A3*, *SPP1* and *DCN*. Collagens are the most abundant proteins in the ECM. Collagen VI is a large, multidomain ECM protein composed of a triple helix of α1, α2, and α3, upregulated in different human cancers that also exhibits a pro-oncogenic role. *COL6A3* expression has been shown to be reduced in obesity, whereas weight loss achieved by caloric restriction and surgery increased *COL6A3* expression in subcutaneous AT [44]. In this sense, MAGP-1 may play a role in OB-associated CC by regulating extracellular matrix composition via COL6A3. Osteopontin, the protein codified by the *SPP1* gene, is an inflammatory factor upregulated in obesity with a critical role in chronic inflammatory diseases and cancer [11]. Together with decorin, the protein codified by *DCN*, both proteins are highly involved in the tumour development process [45,46]. The lack of *DCN* has been associated with the downregulation of E-cadherin and the induction of β-catenin signaling [46]; therefore, MAGP-1 may play a protective role by increasing its gene expression levels. We found a tendency towards increased *MFAP2* gene expression levels after TGF-β treatment in HT-29 cells while stimulation with MAGP-1 enhanced the gene expression levels of *TGFB1*. Microfibrils not only provide structural support to tissues but also control the availability of growth factors, particularly members of the TGF-β superfamily. MAGP-1 sequesters active TGF-β in the ECM and can release its latent form from assembled microfibrils [17]. In this sense, the lack of MAGP-1 would lead to an increase in TGF-β signalling due to the inability of the ECM of the AT to sequester the active form and by the enhanced accumulation of the latent form bounded to microfibrils, creating a larger pool of TGF-β available for activation and signalling. Since MAGP-1 sequesters active TGF-β in the ECM, the increased levels of *TGFB1* after the stimulation with MAGP-1 may constitute a compensatory mechanism with higher concentrations of MAGP-1 being able to reduce the expression of *TGFB1*. *COX2* is commonly upregulated in diverse tumour types, including CC, promoting cell division and proliferation, apoptosis avoidance and angiogenesis [47]. Reportedly, the blockade of the cyclooxygenase (COX) enzyme with nonsteroidal anti-inflammatory drugs (NSAID) has demonstrated an inverse association with the risk of developing CC [48] and human clinical trials have suggested COX-2 selective NSAIDs as a preventive or treatment strategy of CC. In this line, we found that MAGP-1 treatment significantly reduced gene expression levels of *COX2* in HT-29 cells, similarly to anti-inflammatory drugs, supporting the anti-tumoral role of MAGP-1 in CC cells. Several lines of evidence indicate that the extracellular signal-regulated kinase 1/2 (ERK1/2) pathway plays an important role in CC. In this sense, Yao et al. elegantly demonstrated that MAGP-1 activates ERK1/2 through the MFAP2/integrin α5β1/FAK/ERK1/2 pathway in gastric cancer [32]. Moreover, another member of the microfibrillar-associated proteins, MFAP5, has also been implicated in the activation of the ERK signaling pathway in breast cancer cells [49]. In addition, MAGP-1 plays a supportive role in maintaining thermoregulation by indirectly regulating expression of the thermogenic uncoupling proteins (UCPs) [17,22]. In this sense, brain-derived neurotrophic factor (BDNF) also leads to the activation of brown adipose tissue thermogenesis, supporting the potential association between both factors.

The significant decrease in the mRNA levels of *MFAP2* in HT-29 cells preincubated with the adipocyte-derived factors obtained from volunteers with OB strengthens the concept of the interaction between adipocytes and tumoral cells and the contribution of adipocytes promoting a microenvironment favourable to tumour progression probably by inducing inflammation and modifying cancer cell behaviour.

This study has some limitations. Although the number of subjects in the groups may appear somewhat limited, the detailed clinical, biochemical and metabolic characterization of our subjects has to be stressed. Patients were carefully selected to avoid confounding factors and the groups encompassed highly homogeneous patients to perform a robust analysis. The treatment of HT-29 cells with the ACM obtained from patients with obesity and CC would provide important information about the role of AT in the carcinogenesis process. Moreover, the knockdown of *MAFP2* in HT-29 cells would add additional data to confirm the role of MAGP-1 in inflammation and carcinogenesis.

Taken together, the decreased levels of MAGP-1 in patients with obesity and CC as well as its capacity to modulate key genes involved in ECM remodelling and tumorigenesis suggest that MAGP-1 may be an important link between AT excess and CC development (Figure 5). Further studies are warranted to gain further insight into the role of MAGP-1 in the development of obesity-associated CC.

## 4. Material and Methods

### 4.1. Patient Selection

Tissue samples from 79 subjects (29 normoponderal (NP) (17 with CC) and 50 patients with obesity (OB) (19 with CC)) recruited from healthy volunteers and patients attending the Departments of Endocrinology & Nutrition and Surgery at the Clínica Universidad de Navarra were used in the study. Volunteers underwent a clinical assessment including medical history, physical examination and comorbidity evaluation by a multidisciplinary team. Body mass index (BMI) was calculated as weight in kilograms divided by the square of height in meters and body fat (BF) was estimated by the CUN-BAE formula [50]. Patients were classified as NP or with OB according to BMI. Patients suffering from cancer were classified according to the established diagnostic protocol for CC. The pathological characteristics of the subjects with CC included in the study are shown in Appendix A.

VAT samples from NP volunteers were collected from Nissen fundoplication, from subjects with OB from Roux-en-Y gastric bypass (RYGB) and from patients with CC from curative resection for primary colon carcinoma at the Clínica Universidad de Navarra. The control volunteers were healthy, were not on medication and had no signs or clinical symptoms of type 2 diabetes, liver alteration or cancer. The experimental design was approved from an ethical and scientific standpoint (2018.094) for the Hospital’s Ethical Committee responsible for research and all the written informed consents was obtained.

### 4.2. Analytical Procedures

Plasma samples were obtained by venipuncture after an overnight fasting. Glucose was analyzed by an automated analyzer (Hitachi Modular P800, Roche, Basel, Switzerland). Serum concentrations of triglycerides and free fatty acids (FFA) were determined by using commercially available kits (Infinity, Thermo Electron Corporation, Melbourne, Australia). The carcinoembryonic antigen (CEA), fibrinogen and high sensitivity C-reactive protein (CRP) concentrations were measured as previously reported [11]. White blood cell (WBC) count was determined using an automated cell counter (Beckman Coulter, Inc., Fullerton, CA, USA). Circulating levels of MAGP-1, IL-4, IL-6, IL-13, VEGFA, OPN (RayBiotech, Inc., Norcross, GA, USA) and TGF-β (Mybiosource, San Diego, CA, USA) were assessed by commercially available ELISA kits according to the manufacturer’s instructions. The intra- and inter-assay coefficients of variation were <10 and <12% for all analysed molecules.

### 4.3. RNA Extraction and Real-Time PCR

RNA extraction was performed by homogenization with an Ultra-Turrax T25 basic (IKA Werke Gmbh, Staugen, Germany) using QIAzol Reagent (Qiagen, Hilden, Germany) for AT and TRIzol Reagent (Invitrogen, Carlsbad, CA, USA) for HT-29 cells. All samples were treated with DNase I (RNase Free DNase set, Qiagen). Constant amounts of 2 μg of total RNA were reverse transcribed in a 40 μL final volume using random hexamers (Roche) as primers and 400 units of M-MLV reverse transcriptase (Invitrogen, Carlsbad, CA, USA) for cDNA synthesis.

The mRNA levels for *MFAP2*, *TGFB1,* collagen type 6 α3 chain (*COL6A3*), decorin (*DCN*), secreted phosphoprotein 1 (*SPP1*), prostaglandin-endoperoxide synthase 2 (*COX2*/*PTGS2*), mucin 2 (*MUC2*) and catenin β1 (*CTNNB1*) were quantified by Real-Time PCR (7300 Real Time PCR System, Applied Biosystems, Foster City, CA, USA) as previously described [10]. Primers and probes (Appendix A) were designed using the software Primer Express 2.0 (Applied Biosystems) and purchased from Genosys (Sigma-Aldrich, Madrid, Spain). Primers or TaqMan probes encompassing fragments of the areas from the extremes of two exons were designed to ensure the detection of the corresponding transcript avoiding genomic DNA amplification.

The cDNA was amplified using the TaqMan Universal PCR Master Mix (Applied Biosystems) at 95 °C for 10 min, followed by 45 cycles of 15 at 95 °C and 1 min at 59 °C. The primer and probe concentrations were 300 nmol/L and 200 nmol/L, respectively. The *18S* rRNA (Applied Biosystems) was the endogenous control gene for the Real-Time PCR experiments and relative quantification was calculated using the ΔΔCt formula [51]. Relative mRNA expression was expressed as fold expression over the calibrator sample (average of gene expression corresponding to the normoponderal group or unstimulated cells) as previously described [51]. All samples were run in duplicate, and the average values were calculated.

### 4.4. Cell Cultures

The HT-29 cell line, derived from a human colorectal adenocarcinoma, was obtained from ATCC (HTB-38TM, Middlesex, UK) and cultured according to the manufacturer’s instructions. Briefly, cells were seeded at 3 × 10^5^ cells/well and grown in McCoy’s 5A medium with L-glutamine (Sigma) supplemented with 10% fetal bovine serum and antibiotic-antimycotic at 37 °C for 24 h. HT-29 cells were serum-starved for 24 h and then incubated in the presence of tumor necrosis factor (TNF)-α (1, 10, and 100 ng/mL) (Sigma), lipopolysaccharide (LPS) (10, 100 and 1000 ng/mL) (R&D Systems, Minneapolis, MN, USA), interleukin (IL)-4 (1, 10, and 100 ng/mL) (R&D Systems), IL-13 (1, 10, and 100 ng/mL) (R&D Systems), CoCl_2_ (100 and 200 mmol/mL) (R&D Systems), TGF-β (0.1, 1 and 10 ng/mL) (R&D Systems) and adipocyte-conditioned media (ACM) (20% and 40%) for 24 h. In another set of experiments, cells were incubated in the presence of the recombinant form of MAGP-1 (1, 10 and 100 nmol/mL) (R&D Systems) for analysing the expression of related genes.

Human stromavascular fraction cells (SVFC) were isolated from VAT from patients with obesity. SVFC were seeded at 2 × 10^5^ cells/well and grown in adipocyte medium [DMEM/F-12 [1:1] (Invitrogen)] supplemented with 10% newborn calf serum (NCS). After 4 days, the medium was changed to adipocyte medium supplemented with 3% NCS, 0.5 mmol/L 3-isobutyl-1-methylxanthine (IBMX), 0.1 μM dexamethasone, 1 μM BRL49653 and 10 μg/mL insulin. After a 3-day induction period, cells were fed every 2 days with the same medium but without IBMX and BRL49653 supplementation for the remaining 7 days of adipocyte differentiation. The ACM was obtained from these cultures, centrifuged and diluted (20% and 40%).

### 4.5. Statistical Analysis

Data are shown as mean ± standard error of the mean (SEM). CRP concentrations were logarithmically transformed due to their non-normal distribution. The normal distribution of the other variables was adequate for the use of parametric tests. Differences between groups were assessed by two-way ANOVA and one-way ANOVA followed by Tukey’s or Dunnett’s *post-hoc* tests as appropriate. Differences between groups adjusted for age and gender were analyzed by analysis of covariance (ANCOVA). The calculations were performed using the SPSS/Windows version 15.0 statistical package (SPSS, Chicago, IL, USA). A *p* value < 0.05 was considered statistically significant.

## Figures and Tables

**Figure 1 ijms-22-08485-f001:**
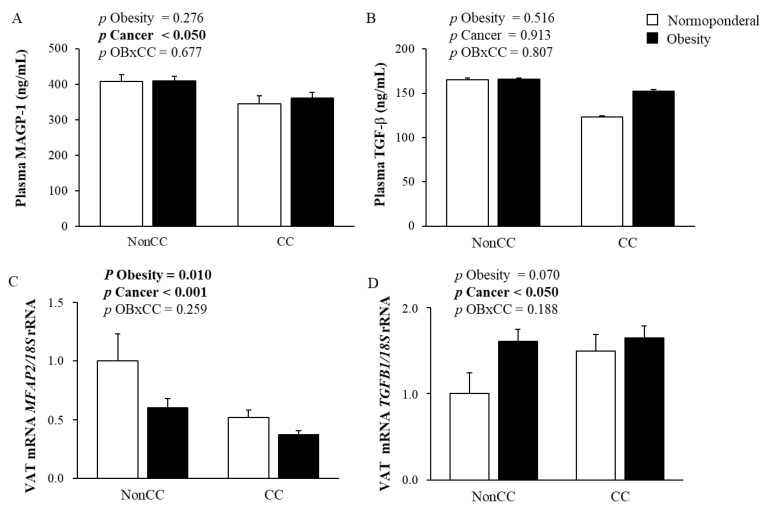
MAGP-1 (**A**) and TGF-β (**B**) circulating levels in normoponderal (NP) volunteers and patients with obesity (OB) with and without colon cancer (CC). *MFAP2* (**C**) and *TGFB1* (**D**) gene expression levels in NP volunteers and patients with OB with and without CC. Differences between groups adjusted for age were analyzed by two-way ANCOVA. Bars represent the mean ± SEM.

**Figure 2 ijms-22-08485-f002:**
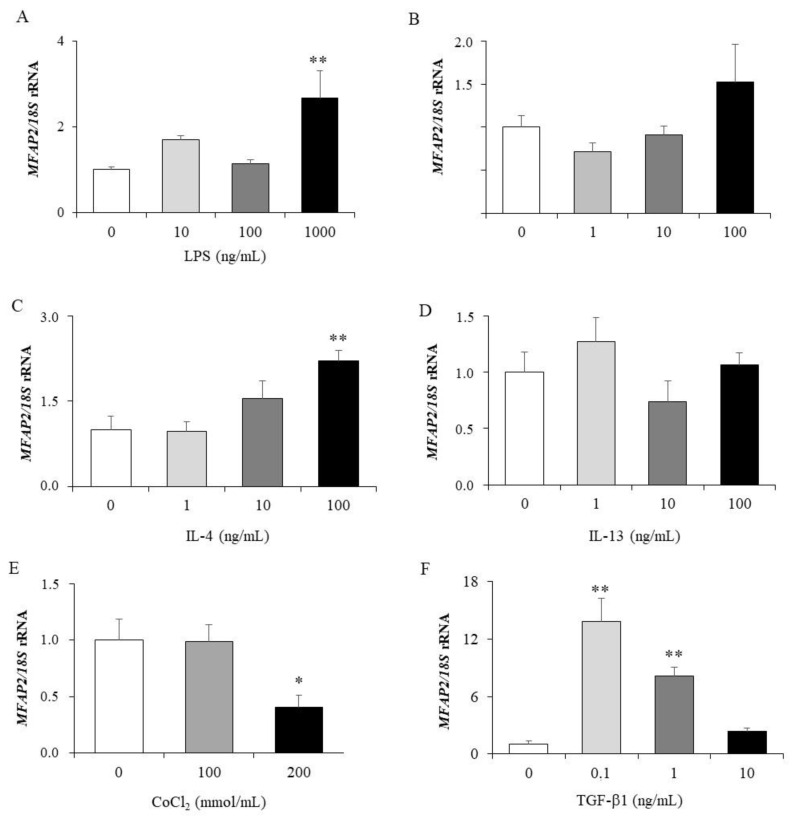
*MFAP2* gene expression levels in HT-29 cells treated with different concentrations of LPS (**A**), TNF-α (**B**), IL-4 (**C**), IL-13, (**D**), CoCl_2_ (**E**) and TGF-β (**F**). Gene expression levels in unstimulated cells were assumed to be 1. Values are the mean ± SEM (*n* = 6 per group). Differences between groups were analysed by one-way ANOVA followed by Dunnett’s *post-hoc* tests. * *p* < 0.05, ** *p* < 0.01 vs. unstimulated cells.

**Figure 3 ijms-22-08485-f003:**
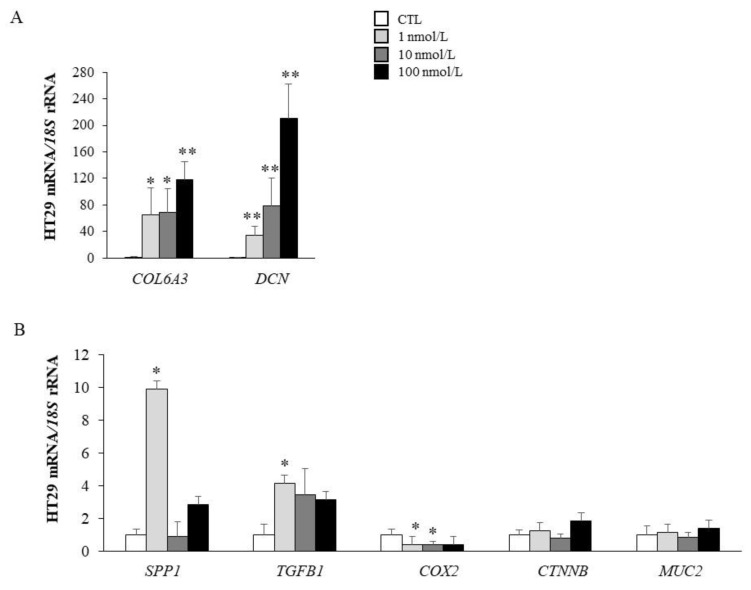
Gene expression levels of extracellular matrix remodelling markers (**A**) as well as tumorigenesis-related molecules (**B**) in HT-29 cells stimulated with different concentrations of recombinant MAGP-1 for 24 h. Gene expression levels in unstimulated cells were assumed to be 1. Values are the mean ± SEM (*n* = 6 per group). Differences between groups were analyzed by one-way ANOVA followed by Dunnett’s *post-hoc* tests. * *p* < 0.05 and ** *p* < 0.01 vs. unstimulated cells.

**Figure 4 ijms-22-08485-f004:**
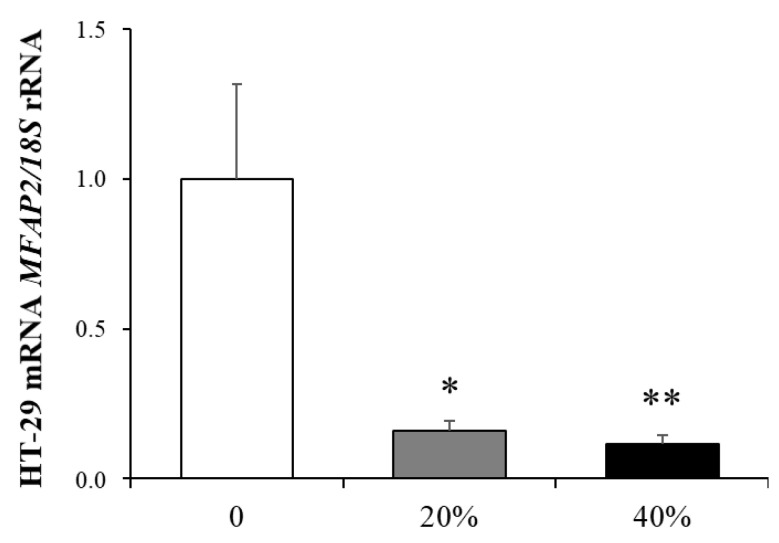
Bar graphs show the effect of ACM (20% and 40%) from obese subjects on the transcript levels of *MFAP2* in HT-29 cells. Values are the mean ± SEM (*n* = 6 per group). Differences between groups were analyzed by one-way ANOVA. * *p* < 0.05 and ** *p* < 0.01 vs. unstimulated cells.

**Figure 5 ijms-22-08485-f005:**
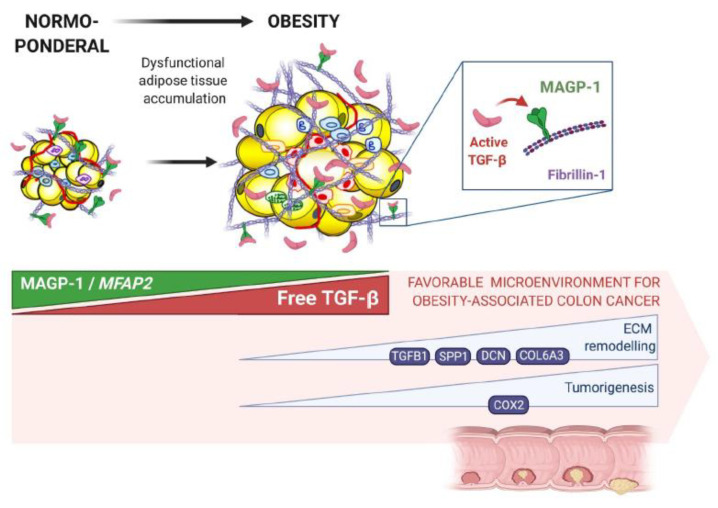
Proposed mechanism of the involvement of MAGP-1 in obesity-associated CC development. Decreased *MFAP2* gene expression levels in the visceral adipose tissue from patients with obesity and colon cancer may be responsible for the increased free active TGF-β enhancing its downstream signalling pathways associated with inflammation, ECM remodelling and tumorigenesis. Therefore, MAGP-1 may participate in creating a favourable microenvironment to tumorigenesis in obesity and, thus, function as a link between adipose tissue excess and obesity-associated CC development.

**Table 1 ijms-22-08485-t001:** Anthropometric and biochemical characteristics of the subjects included in the study.

	Normoponderal		Obesity		*p* OB	*p* CC	*p* OBxCC
	Non-Colon Cancer	Colon Cancer	Non-Colon Cancer	Colon Cancer			
*n* (male, female)	12 (5, 7)	17 (8, 7)	31 (18, 13)	19 (14, 5)			
Age (years)	53 ± 2	63 ± 3	55 ± 1	64 ± 3	0.518	<0.001	0.925
Body weight (kg)	62.7 ± 1.9	61.2 ± 5.1	83.2 ± 2.0	78.3 ± 2.2	<0.001	0.577	0.320
Body mass index (kg/m^2^)	22.7 ± 0.9	22.4 ± 0.4	30.2 ± 0.7	29.6 ± 0.7	<0.001	0.109	0.588
Estimated body fat (%)	29.9 ± 1.9	29.2 ± 1.5	37.4 ± 1.1	33.4 ± 1.4	<0.001	0.155	0.370
Waist (cm)	83 ± 1	80 ± 1	99 ± 2	111 ± 2	<0.001	0.241	0.180
Fasting glucose (mg/dL)	102 ± 4	141 ± 13	110 ± 5	128 ± 8	0.730	<0.001	0.143
Free fatty acids (mg/dL)	12.7 ± 1.4	26.5 ± 2.4	15.4 ± 1.2	22.2 ± 1.7	0.570	<0.001	0.064
Triglycerides (mg/dL)	87 ± 10	112 ± 11	117 ± 9	121 ± 20	0.747	0.752	0.685
C-reactive protein (mg/L)	0.20 ± 0.13	1.10 ± 0.96	1.17 ± 0.08	8.48 ± 1.84 ***	0.008	<0.001	0.031
Fibrinogen (mg/dL)	337 ± 27	277 ± 26	300 ± 17	451 ± 30 **	0.159	0.418	0.033
IL-4 (pg/mL)	11.03 ± 1.70	9.22 ± 0.31	9.13 ± 0.21	8.64 ± 0.17	0.038	0.046	0.315
IL-6 (pg/mL)	3.34 ± 0.26	4.45 ± 0.45	5.23 ± 1.11	9.26 ± 1.34	0.002	<0.001	0.123
IL-13 (pg/mL)	0.71 ± 0.06	0.79 ± 0.09	0.75 ± 0.03	0.57 ± 0.03	0.016	0.900	0.216
OPN (ng/mL)	25.30 ± 3.30	38.93 ± 4.48	28.76 ± 2.42	70.15 ± 10.13	<0.001	<0.001	0.098
VEGF (ng/mL)	16.01 ± 0.85	16.31 ± 0.70	19.26 ± 0.95	18.39 ± 1.27	0.038	0.926	0.682
YKL-40 (ng/mL)	27.00 ± 2.19	37.46 ± 8.62	39.20 ± 4.17	63.57 ± 7.72	0.006	0.012	0.309
CEA (ng/mL)	1.58 ± 0.32	2.55 ± 0.44	1.68 ± 0.28	8.41 ± 2.60	0.267	0.021	0.401
Leucocyte (×10^9^/L)	6.17 ± 0.91	8.19 ± 1.14	6.22 ± 0.31	7.73 ± 0.83	0.823	0.024	0.653

Data are mean ± SEM. CEA, carcinoembryonic antigen; CC, colon cancer; IL, interleukin; NP, normoponderal; OB, obesity, OPN, osteopontin; VEGF, vascular endothelial growth factor; YKL-40, chitinase 3-like 1. Statistical differences were analyzed by two-way ANCOVA and one-way ANCOVA followed by Tukey’s *post-hoc* tests as appropriate. ** *p* < 0.01, *** *p* < 0.001 vs. NP-non CC.

## Data Availability

The data presented in this study are available on request from the corresponding author upon reasonable request.

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
