# Peer review of "Decreased Levels of Microfibril-Associated Glycoprotein (MAGP)-1 in Patients with Colon Cancer and Obesity Are Associated with Changes in Extracellular Matrix Remodelling"

_ijms, 2021, doi:10.3390/ijms22168485_

Round 1

Reviewer 1 Report

Authors are requested to present a valid and succinct conceptual frame work in the manuscript and recent references must be cited

Author Response

We are very grateful for the comments of the Reviewer, which have served to substantially improve the manuscript.

1.     Authors are requested to present a valid and succinct conceptual frame work in the manuscript and recent references must be cited.

Following the Reviewer’s comment, a concise framework has been incorporated to the Introduction in the revised version of the manuscript.

We also agree with the Reviewer’s comment and according to his/her suggestion recent references have been cited in the revised version of the manuscript.

Reviewer 2 Report

Review –

The authors present an interesting finding related to the expression of MAGP-1 and colon cancer. While the specific risk of obesity is highly correlated with colon cancer, the role of MAGP-1 is less well understood.

  1. Some minor comments regarding gene nomenclatures-

“Obesity and colon cancer decrease circulating and gene levels of MAGP-1”. This sentence is not correct, as it implies circulating DNA levels we quantified. It is clear from the methods that ELISAs were used to assess MAGP-1. The gene name at the DNA levels is nomenclated differently than the resultant protein. Please change throughout the manuscript.

  1. (Table 1) One of key findings of the study is that c-reactive protein and fibrinogen, are significantly elevated in obese CC patients as compared to non-obese CC patients. Could the authors further measure or indicate additional markers of early onset tissue damage via fibrogenesis, which is presumably occurring in obese patients concomitantly in cancer development.
    1. Furthermore, is there any indication obese patients are more/less likely to diagnose with early/late stage diseases as compared to non-obese cancer patients.
  2. (Figure 1). There is no indication MAGP-1 levels (DNA or protein) are varied in non-obese vs obese colon cancer patients. Can the authors clarify/rectify this concern? Perhaps there is a non-TGFB pathway activated herein?
  3. (Figure 2). Interesting for the authors to treat HT-29 cells with substances such as LPS, TNFa/b, and IL’s; yet, no to measure any of these components in human specimens (Table 1). Inflammatory markers can be reliably measured, as well as particular inorganic salt levels, and should be performed here.
  4. (Figure 3) is unclear. This figure also brings up concerns regarding consistency of the work. For instance, COL6A3, DCN, and SPP1 were measured after treatment of MAGP1, but after treatments with LPS or ILs/TGFs… This probably should be done.
    1. Also, do MAGP-1 KD cells (in HT-29) behave differently in their response to immune stimuli and control HT-29 cells.?
  5. (Figure 4) is unclear. Adding ACM (20% and 40%) from obese vs obese cancer patients would provide relevant information
  6. (Figure 5). Figure is unclear and mentions concepts such as COX2, which was not discussed herein. Overall SPP1 is known to be associated with obesity and cancer risk, and the authors should determine if the BDNF pathway, ERK pathway, and/or the Early T-lymphocyte activation pathway are affected by MAGP1. This is because there is already a strong correlation between early immune active, cancer activation, and cancer metastasis in cancer such as Head and neck cancer. These pathways could paly a role in colon cancer and should be addressed in this manuscript.

Reviewer 3 Report

The manuscript is well-written and describes a novel and interesting study. However, the research design has some flaws.

Major points:

  1. The study population rise some concerns namely the reduced number of recruited participants, the fact that NP volunteers are not age-match with patients, and also the absence of gender analysis. Several studies showed that gender and age influence a variety of adipose tissue biology including adipocyte function, sex hormone effects, genetics, and metabolic inflammation. So, gender and age cannot be ignored, and NP and patients need to be age and gender-matched.
  2. The Authors also need to explain why they state that “Gene expression levels in unstimulated cells were assumed to be 1” (Figure 2 and 3 captions). According to the Material and Methods section, the Authors used the 2^DDCt formula. This formula gives the gene expression fold change (is a normalization to control), and the control sample will be 1 and do not have standard deviation or SEM. The vertical legend of graphs also needs to be revised accordingly.
  3. The influence of pathological features of CC in MAGP-1/MFAP2 and TGF-b1/TGFB1 expression should be explored.
  4. The Author should include a reflection on the limitations of the study in the Discussion section.

Minor points:

  1. In table 1, the P OB, P CC, and P OBxCC abbreviations should be written out in full in the table caption. 
  2. In line 30, TNS-a should be replaced with TNF-a.

Reviewer 4 Report

Report on the manuscript Decreased levels of microfibril-associated glycoprotein (MAGP)-1 in patients with colon cancer and obesity are associated with changes in extracellular matrix remodeling by Iranzu Gomez de Segura et al.

In the paper, the authors present results suggesting that (MAGP)-1 downregulation accompanied with TGF-β overexpression is responsible for local inflammation and ECM remodeling in obesity-related colon cancer development. This is assumed from the gene expression analysis of tissue samples taken from healthy people and patients suffering from obesity (OB), colon cancer (CC), and obesity accompanied with CC. Additionally, the hypothesis has been tested with experiments performed on the immortalized colon cancer cell line HT-29. The paper presents valuable findings. Yet, in my opinion, there are some shortcomings the authors should refer to.

  1. My major remark regards tissue samples analysis. Data presented in the paper are limited to MFAP2 and TGF – β1 gene expression. If the aim of the study is to show the influence of these factors on ECM remodeling, the analysis of the ECM composition of the studied tissues should be performed (collagen 6 levels for example) and compared between the groups. This lack is highlighted when a reader compares tissue data to HT-29 results.
  2. In Figure 1, the way of presentation of the data and their statistical analysis should be changed. Neither the legend nor the figure capture allows the reader to get which groups are compared and where the statistically significant differences were found. Y-axis in Figure 1C seems to be mislabeled.
  3. Description and discussion of data presented in Figure 3 are dissonant. In line 41, the authors claim that ‘… MAGP-1 treatment reduced gene expression levels of PTGS2.’, while there is no such data in the figure. On the other hand, in the figure data regarding levels of COX2 are shown, while the authors do not refer to it in the text.
  4. In Figure 3, it is shown that MAGP-1 treatment increased the TGF– β1 level. Statistically, a significant difference is depicted only for 1nmol/mL concentration of MAGP-1. The investigation of the figure also indicates that with the increase of MAGP-1 concentration, the TGF – β1 level decreases. The authors do not comment on that.
  5. In the caption of figure 3, the authors may want to use (a) and (b) cross-references.
  6. Please check the caption of figure 4 (line 57).
  7. In the paper, the authors use both TGFB1 and TGF – β1. Please unify.

Round 2

Reviewer 2 Report

The reviewer appreciations the corrections made in the manuscript.

I am aware this sentence structure is wrong, which is why i flagged it.

..circulating and gene levels of MAGP-1...

Yes the correct sentence structure is

...circulating and gene expression levels of MAGP-1...

Also the reviewer appreciates the transparency in explaining the datasets.  The additional information now provided regarding ILs included in the analysis (illustrated in the table) address many of the reviewer concerns.

Reviewer 3 Report

The authors answered the reviewers' comments accordingly.  I have no further comments.

Reviewer 4 Report

I wish to thank the authors for their responses and changes induced in the paper. I have no further comments.